# Factors associated with mortality of the elderly due to ambulatory care sensitive conditions, between 2008 and 2018, in the Federal District, Brazil

**Geraldo Marques da Costa** [ID][☯]*, **Mauro Niskier Sanchez**[☯], **Helena Eri Shimizu**[☯]

Public Health Department, Faculty of Health Sciences, University of Brasilia, Brasilia, Federal District, Brazil

☯ These authors contributed equally to this work.
* geraldomarques06@gmail.com

**Data Availability Statement:** All relevant data are within the manuscript and its Supporting information files.

## Abstract

### Introduction

In Brazil, the Unified Health System (Sistema Único de Saúde, or SUS) provides health care, and an aging population overwhelms the system due to the greater vulnerability of the elderly. In the Federal District, two models of primary care coexist–the traditional primary care and the family health strategy. The present study aimed to analyze the factors associated with mortality of the elderly due to conditions sensitive to ambulatory care in the Federal District, Brazil.

### Materials and methods

This cross-sectional study investigated all deaths that occurred in people over 60 years old between 2008 and 2018. The variables studied were age at death, sex, marital status, education, race/color, death by condition sensitive to ambulatory care, and population coverage of primary care services. The Urban Well-Being Index (UWBI) was used, which includes the dimensions: mobility, environmental and housing conditions, infrastructure, and collective services, to analyze issues related to the place where the senior citizen resides.

### Results

The deaths 70,503 senior citizens were recorded during the study period. The factors associated with mortality in the elderly due to ambulatory care sensitive conditions were male, lower income, and less education. Residing in a place with poor UWBI presented a response gradient with higher mortality. Increased ambulatory care coverage was also associated with lower mortality.

### Conclusions

The study evidenced an association between male gender, age, income, and education, and UWBI with lower mortality due to ambulatory care sensitive conditions, and these

**Funding:** The author(s) received no specific funding for this swork.

**Competing interests:** The authors have declared that no competing interests exist.

associations presented a response gradient. The study also found that increased coverage of the elderly population was associated with lower mortality from sensitive conditions.

## Introduction

Due to the improved life expectancy, the increased number of senior citizens has pressured health systems for more efficient responses. In Brazil, the Unified Health System (Sistema Único de Saúde, or SUS) is tasked to guarantee the right to health care, through health promotion, protection, and recovery procedures, considering the social determinants of health. The strong and growing link between socio-economic aspects and health continues to increase, particularly in the elderly population that has diverse vulnerabilities [1].

Good primary health care (PHC) helps to reduce hospitalizations and can impact the mortality of the elderly [2]. PHC requires qualified professionals, service structure, and coordination with other levels of services.

In Brazil, PHC is configured in two models: the traditional primary care (TPC) consisting of specialist physicians (clinician, gynecologist, and pediatrician) and the Family Health Strategy (FHS) formed by a team with a doctor, nurse, nursing technician, and community health agent who works in the area with a registered population of at most 4,000 people [3].

Studies have shown that coverage by primary care contributes to the reduction of mortality in the population, including the elderly [4]. Mortality from ambulatory care sensitive causes is an important performance indicator [5]. In Brazil, Ordinance 221/2008 of the Ministry of Health created a list of avoidable hospitalizations by effective and accessible ambulatory care services to assess the quality of care provided; however, PHC remains scantly available to the elderly population [6]. Different countries have their own lists of ambulatory care sensitive conditions. A review showed a strong association between low socio-economic status and risk of hospitalization for sensitive conditions, thus the relationship between ambulatory care and population morbidity is internationally evidenced [7].

It is known that other general socio-economic, cultural, and environmental factors influence the risk of dying in groups according to the levels of income, education, profession, sex, place of residence, and other factors related to mortality in the elderly [8]. Worsening poverty and social exclusion have become increasingly evident since 2016 with economic austerity policies, especially in large urban centers, with greater social inequalities, increasing extreme poverty and hunger [9]. The issue of decent housing has been understood as a determinant of health and well-being, linked to patterns of illness, especially in communicable diseases, domestic accidents, and mental health [10].

Therefore, analysis in greater depth is needed on how these factors influence the mortality of the elderly due to ambulatory care sensitive conditions. This situation still needs further elucidation in the literature.

We hope that this study will increase understand about the influence of housing conditions and urban infrastructure, measured by the Urban Well-Being Index (UWBI), and primary care coverage on the mortality of the elderly. The hypothesis is that greater primary care coverage and better housing conditions can impact mortality by reducing deaths from conditions sensitive to ambulatory care.

This study analyzed the factors associated with mortality in the elderly due to ambulatory care sensitive conditions in the Federal District, Brazil, from 2008 to 2018.

## Materials and methods

The study is based on secondary data obtained from the database of the Health Department of the Federal District. Confidentiality and anonymity were observed in accordance with the Resolution of the National Health Council (CNS) n˚ 466, of December 12, 2012. The project was approved by the Research Ethics Committee of the Faculty of Health at the University of Brasília, opinion n˚ 3,479,132; issued on July 31, 2019, Certificate of Presentation for Ethical Appraisal (CAAE) No. 08615719.3.0000.0030. The ethics committee waived the requirement for informed consent.

This is a cross-sectional study, based on mortality data of senior citizens over 60 years old, from 2008 to 2018 in the Federal District, Brazil. In Brazil, a senior citizen is defined as an individual aged 60 years or over [11].

Data were extracted from the mortality information system of the Health Department of the Federal District. This system is based on death certificates and is coordinated by Epidemiological Surveillance. The data are monitored and when inconsistencies are found, additional information is requested. Technicians monitor the information to ensure data quality. In addition, the Federal District has a Death Verification Service to assess deaths whose causes are not well elucidated.

The Federal District is in the Central Plateau of Brazil, located in the Center-Western region. The population in 2019 was estimated at 3,012,718, of which 328,379 were senior citizens [12]. The population studied consisted of all deaths that occurred in the Federal District in people 60 years old or more, between the years 2008 to 2018. Records referring to senior citizens who did not reside in the Federal District, or those with incomplete records were excluded.

Information on coverage of PHC was obtained from the Ministry of Health's website for information and management of primary care is www.egestorab.saude.gov.br. Coverage data were standardized using as a reference December of each year of the analyzed historical series from 2008–2018.

The variables studied were: age at death (60–69, 70–79, >80), sex (male and female), marital status (single, married, widowed, separated/divorced, not stated), education (none, 0–3, 4–7, 8–11, >12 years of schooling), race/color (white, black, brown, Asian, indigenous, not stated) death by ambulatory care sensitive conditions, coverage by TPC and FHS. In other words, the deceased elderly person lived in a household attended by Primary Care in the traditional model or in the family health strategy model in Federal District.

The variable used to analyze housing and urban infrastructure issues was the UWBI, considering the location where the elderly person resided in Federal District. The UWBI is calculated by the address and consists of five dimensions: mobility, environmental conditions, housing conditions, infrastructure, and collective services. Each of these dimensions is calculated, and then the arithmetic mean is calculated. The result is a value ranging from 0 to 1, in which the UWBI is better the closer to 1 and worse the closer to 0 [13].

The diseases were classified as ambulatory care sensitive conditions according to the ordinance 221/2008 of the Ministry of Health [14]. The tenth version of the International Classification of Diseases (ICD), known as the ICD-10, was used.

The statistical analysis of the data initially relied on the bivariate analysis of the variables of interest to study the outcome (death by sensitive condition). In the next step, a multivariate logistic regression analysis was performed [15], which was built according to a hierarchical structure of three blocks (distal, medial, and proximal). The adjustment variables were sex and age group for the model as a whole but were not restricted to just one of the considered levels.

The first block (distal) included the income group, which also represents a regional characterization for the regions in the Federal District. The second block (medial) had the expression

variables of urban well-being, expressed by the UWBI composite index, together with coverage of primary care (TPC) and family health teams (FHS). The third block contained education, marital status, and race/color, which are proximal characteristics.

Statistical analyzes of data were performed using the R [16] v program. 2.2.5019 R Core Team, 2019.

## Results

There were 70,503 deaths among the elderly living in the Federal District between 2008 and 2018 (Table 1). Most deaths occurred in women (50.2%), with older seniors predominating (40.7%). Married individuals (38.4%) predominated, followed by widowers (31,3%). Most deaths occurred in seniors who were declared white (55,1%). A significant percentage of deaths resulted from ambulatory care sensitive conditions (29.2%).

In Table 2, deaths in the elderly due to ambulatory care sensitive conditions were correlated with several factors. Men had a higher risk of dying from sensitive conditions. The data on the age group, adjusted for age, elucidated a gradient of greater risk of dying as the seniors' age increased. Adjusted for sex and age, seniors with lower income exhibited a higher risk of death, but lower middle income had a slightly higher risk of dying than low income.

For the UWBI, when adjusted for sex, age, and income, a dose-response relationship was observed–the worse the UWBI, the greater the risk of the senior dying due to sensitive conditions. An even lower risk of dying was noted related to primary care coverage, both in traditional TPC and in the FHS. The race/color of the senior, adjusted for sex, age, income, UWBI, and primary care coverage, did not obtain statistical significance in terms of the risk of dying. In the analysis of marital status, being married had a protective effect on the risk of dying for the elderly from sensitive conditions. Education had a dose-response effect–the lower the education level, the higher the risk of dying from the studied causes.

**Table 1. Distribution of deaths of the elderly according to sociodemographic characteristics and ambulatory care sensitive conditions.** Federal District, Brazil, 2008 to 2018.

| | | N° of deaths | % |
|---|---|---|---|
| Sex | Female | 35.402 | 50.2 |
| | Male | 35.101 | 49.8 |
| Age group | 60 to 69 years old | 18.993 | 26.9 |
| | 70 to 79 years old | 22.820 | 32.4 |
| | 80 years old or above | 28.690 | 40.7 |
| Marital status | Single | 12.467 | 17.7 |
| | Married | 27.105 | 38.4 |
| | Widowed | 22.052 | 31.3 |
| | Separated/divorced | 6.920 | 9.8 |
| | Not indicated | 1.959 | 2.8 |
| Race/color | White | 38.816 | 55.1 |
| | Black | 4.520 | 6.4 |
| | Asian | 437 | 0.6 |
| | Brown | 26.179 | 37.1 |
| | Indigenous | 43 | 0.1 |
| | Not indicated | 508 | 0.7 |
| Sensitive Condition | Yes | 20.606 | 29.2 |
| | No | 49.897 | 70.8 |

Source: Database of the Federal District Health Department

**Table 2. Hierarchical model: Factors associated with deaths of the elderly from ambulatory care sensitive conditions, Distrito Federal, Brazil. 2008–2018.**

| Independent variables | | Total | Sensitive condition | | Model A (Block 1) | | | | Model B (Block 2) | | | | Model C (Block 3) | | | |
|---|---|---|---|---|---|---|---|---|---|---|---|---|---|---|---|---|
| | | | Yes | No | Odds ratio | 95% confidence interval | | P-value | Odds ratio | 95% confidence interval | | P-valor | Odds ratio | 95% confidence interval | | P-valor |
| | | | | | | Upper limit | Lower limit | | | Upper limit | Lower limit | | | Upper Limit | Lower limit | |
| **Sex** | Female | 35402 | 10953 | 24449 | 1.133 | 1.094 | 1.173 | <0.001 | 1.135 | 1.095 | 1.175 | <0.001 | 1.051 | 1.011 | 1.092 | 0.012 |
| | Male (ref.) | 35101 | 9653 | 25448 | | | | | | | | | | | | |
| **Age group (years)** | 60 to 69 (ref.) | 18993 | 4431 | 14562 | | | | | | | | | | | | |
| | 70 to 79 | 22820 | 6615 | 16205 | 1.378 | 1.314 | 1.444 | <0.001 | 1.380 | 1.316 | 1.447 | <0.001 | 1.332 | 1.270 | 1.398 | <0.001 |
| | 80 and over | 28690 | 9560 | 19130 | 1.775 | 1.696 | 1.857 | <0.001 | 1.794 | 1.714 | 1.877 | <0.001 | 1.650 | 1.571 | 1.733 | <0.001 |
| **Income group** | High (ref.) | 13003 | 3010 | 9993 | | | | | | | | | | | | |
| | Upper middle | 27206 | 7886 | 19320 | 1.435 | 1.363 | 1.511 | <0.001 | | | | | | | | |
| | Lower middle | 26075 | 8387 | 17688 | 1.762 | 1.672 | 1.856 | <0.001 | | | | | | | | |
| | Lower | 3935 | 1235 | 2700 | 1.748 | 1.605 | 1.903 | <0.001 | | | | | | | | |
| **UWBI*** | Very good (ref.) | 11605 | 2654 | 8951 | | | | | | | | | | | | |
| | Good | 8951 | 2355 | 6596 | | | | | 1.148 | 1.052 | 1.254 | 0.002 | | | | |
| | Average | 25434 | 7689 | 17745 | | | | | 1.341 | 1.216 | 1.480 | <0.001 | | | | |
| | Bad | 23906 | 7718 | 16188 | | | | | 1.429 | 1.269 | 1.609 | <0.001 | | | | |
| | Terrible | 323 | 102 | 221 | | | | | 1.550 | 1.168 | 2.057 | 0.002 | | | | |
| **FHS Coverage*** | % | | | | | | | | 0.995 | 0.991 | 0.999 | 0.011 | | | | |
| **TPC Coverage***** | % | | | | | | | | 0.996 | 0.994 | 0.999 | 0.001 | | | | |
| **Race/color** | White (ref.) | 38816 | 11045 | 27771 | | | | | | | | | | | | |
| | Black | 4520 | 1440 | 3080 | | | | | | | | | 1.067 | 0.992 | 1.148 | 0.080 |
| | Brown | 26179 | 7833 | 18346 | | | | | | | | | 1.007 | 0.968 | 1.047 | 0.736 |
| | Others | 480 | 149 | 331 | | | | | | | | | 1.048 | 0.841 | 1.304 | 0.678 |
| **Marital status** | Single (ref.) | 12467 | 3923 | 8544 | | | | | | | | | | | | |
| | Married | 27105 | 7045 | 20060 | | | | | | | | | 0.862 | 0.819 | 0.908 | <0.001 |
| | Widowed | 22052 | 7209 | 14843 | | | | | | | | | 0.993 | 0.943 | 1.046 | 0.786 |
| | Separated/ legally divorced | 6920 | 1858 | 5062 | | | | | | | | | 0.945 | 0.880 | 1.014 | 0.114 |
| **Education** | None | 13848 | 4912 | 8936 | | | | | | | | | 1.465 | 1.354 | 1.584 | <0.001 |
| | 1 to 3 years | 20462 | 6345 | 14117 | | | | | | | | | 1.324 | 1.232 | 1.423 | <0.001 |
| | 4 to 7 years | 11638 | 3286 | 8352 | | | | | | | | | 1.216 | 1.128 | 1.311 | <0.001 |
| | 8 to 11 years | 10255 | 2527 | 7728 | | | | | | | | | 1.134 | 1.053 | 1.221 | 0.001 |
| | 12 years or more (ref.) | 7922 | 1627 | 6295 | | | | | | | | | | | | |

* UWBI: urban well-being index

** FHS coverage: family health strategy coverage

*** TPC coverage: traditional practice care coverage.

## Discussion

The present study found that being older and being male were factors associated with the mortality of senior citizens due to conditions sensitive to ambulatory care in the Federal District. These findings corroborate the results of other studies that showed a higher risk of dying among elderly men, who tend to have worse health indicators, probably due to their low adherence to preventive measures, and the consequent reduction in autonomy for activities of daily living and less involvement with community activities [17–19].

When the marital status of the senior citizens who passed away was analyzed, being married seemed to have a protective effect against death due to conditions sensitive to ambulatory care. Living with a spouse reduces the mortality of the elderly [20]. This effect can be explained by better financial conditions, better education, and premarital conditions observed in married elderly [21].

Income conditions increased the chance of seniors dying from conditions sensitive to ambulatory care. Poverty, as well as the deprivations resulting from it, has worsened in recent years, compromising the survival of the elderly [18]. In Japan, this situation was related to thousands of premature deaths [22]. In the state of Pará, a study observed that the purchasing power of the elderly as well as access to work and income were associated with greater survival of senior citizens and should be observed and promoted by public health policies [23].

A gradient effect was observed, in that mortality of the elderly due to sensitive conditions decreased inversely proportional to educational level. Higher education levels contribute to better self-care in the elderly, in addition to facilitating therapeutic adherence and understanding of health guidelines [24].

For the UWBI, when adjusted for sex, age, and income, a dose-response effect was observed in this study. The worse their UWBI, the greater their chance of dying from sensitive conditions. The data indicate that the great regional inequalities in the Federal District affect the elderly, which include poor housing, mobility, and environmental conditions as well as collective services. A study of the Federal District showed that the UWBI ranged from 0.26 to 0.97, and 19% of the regions had poor UVBI, while 6% were classified as very poor [13]. Senior citizens in these regions live in very precarious conditions, in informal housing, on unpaved streets, without sewage and running water, and with pollution. As shown by other studies, these conditions lead to several diseases, increasing the early mortality of the elderly [25–27].

The coverage of the population by primary care exhibited a protective effect on the mortality of the elderly due to sensitive conditions. The Federal District has had a significant increase in the number of family health teams in recent years, especially through the 2017 policy of converting traditional TPC teams to FHS [28]. In addition to increasing coverage, this policy focused on improving the care model, through investments in professional training and monitoring [29] and greater capacity to offer comprehensive and holistic care to all segments of the population, including the elderly. The family health teams benefited from the *Programa Mais Médico* (more doctors), which alleviated the problem of the chronic lack of doctors in peripheral regions and other areas with severe poverty [30].

Several studies reiterate that the good coverage of primary care facilitates access to medical appointments, complementary exams, and the dispensing of medications, which helps reduce mortality of the vulnerable elderly population, especially due to complications of chronic diseases [31–33].

Mortality studies can be impacted by the quality reporting of the causes of death. As this is an ecological study, an ecological bias is possible for all the studied relationships. This study analyzed proportional mortality, which may limit observations on risk of dying; however, it permits conclusions about the chance of association with mortality. Furthermore, the

modeling of this study did not permit certification that the deceased elderly person used the PHC service. In addition, income and UWBI were estimated based on the place of residence, which may not correspond to the individual reality of the senior citizens investigated. Another limiting factor for the analysis was the change in the primary care model in 2017, which occurred near the end of the period studied.

## Conclusions

The study evidenced an association between the sex, age, income, and education with mortality from ambulatory care sensitive conditions, with the strongest association between poor UWBI and the chance of dying in the elderly. These associations showed a response gradient. This finding indicates the need for public investments to improve urban living conditions to help reduce the death rate of this population. The study also found that increased PHC coverage of the elderly population was associated with lower mortality from sensitive conditions. Therefore, public policies to expand and strengthen primary care have the potential to protect and prolong the lives of the elderly.

## Supporting information

**S1 Data.**
(CSV)

**S2 Data.**
(XLSX)

## Author Contributions

**Conceptualization:** Geraldo Marques da Costa, Mauro Niskier Sanchez, Helena Eri Shimizu.

**Data curation:** Geraldo Marques da Costa, Helena Eri Shimizu.

**Formal analysis:** Geraldo Marques da Costa, Mauro Niskier Sanchez, Helena Eri Shimizu.

**Investigation:** Geraldo Marques da Costa, Helena Eri Shimizu.

**Methodology:** Geraldo Marques da Costa, Mauro Niskier Sanchez, Helena Eri Shimizu.

**Project administration:** Geraldo Marques da Costa, Helena Eri Shimizu.

**Validation:** Geraldo Marques da Costa, Helena Eri Shimizu.

**Writing – original draft:** Geraldo Marques da Costa, Helena Eri Shimizu.

**Writing – review & editing:** Geraldo Marques da Costa, Mauro Niskier Sanchez, Helena Eri Shimizu.

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
