## [Decision Letter · Decision Letter 0]

11 Apr 2022

PONE-D-21-36313FACTORS ASSOCIATED WITH MORTALITY OF THE ELDERLY DUE TO AMBULATORY CARE SENSITIVE CONDITIONS, BETWEEN 2008 AND 2018, IN THE FEDERAL DISTRICT, BRAZILPLOS ONE

Dear Dr. Marques da Costa,

Thank you for submitting your manuscript to PLOS ONE. After careful consideration, we feel that it has merit but does not fully meet PLOS ONE’s publication criteria as it currently stands. Therefore, we invite you to submit a revised version of the manuscript that addresses the points raised during the review process. I acted as the academic editor and one of the reviewers. I invited several reviewers, without luck or they could not return the manuscript on time. This is a very interesting paper and contributes to the discussion of health-mortality and SES. In addition to comments by reviewer # 1, I have some comments and suggestions The paper could provide more detailed information on the quality of information and how it changes over the period of analysis. Is the quality of information similar in 2008 and 2018?The methodology could provide more detail to the reader. And I would like to see some theoretical discussion on the choice of variables and the expected results.Coverage and urban well-being do not vary by individual, they only vary by period of time. What is the limitation of doing this? Does it act as a control for period? Mortality increases with age - there are more deaths at older ages than any other age. Thus, it is expect that we see deaths increasing with your age in the analysis. One alternative is to model age specific mortality rates. One can obtain population by sex and age from IBGE or other sources. Then, you could model death rates or apply a poisson model on the counts of death. I believe modeling only death counts without a exposure measure limits the discussion of the results and might be affecting your results. most of your discussion and conclusions talk about risk. But your analysis is not about rates or risks. This should be revised or additional analysis performed. Please submit your revised manuscript by May 26 2022 11:59PM. If you will need more time than this to complete your revisions, please reply to this message or contact the journal office at plosone@plos.org. Please include the following items when submitting your revised manuscript:A rebuttal letter that responds to each point raised by the academic editor and reviewer(s). You should upload this letter as a separate file labeled 'Response to Reviewers'.A marked-up copy of your manuscript that highlights changes made to the original version. You should upload this as a separate file labeled 'Revised Manuscript with Track Changes'.An unmarked version of your revised paper without tracked changes. You should upload this as a separate file labeled 'Manuscript'.

We look forward to receiving your revised manuscript.

Kind regards,

Bernardo Lanza Queiroz, Ph.D

Academic Editor

PLOS ONE

Reviewers' comments:

Reviewer's Responses to Questions

**Comments to the Author**

1. Is the manuscript technically sound, and do the data support the conclusions?

Reviewer #1: Partly

2. Has the statistical analysis been performed appropriately and rigorously? 

Reviewer #1: Yes

3. Have the authors made all data underlying the findings in their manuscript fully available?

Reviewer #1: Yes

4. Is the manuscript presented in an intelligible fashion and written in standard English?

Reviewer #1: Yes

5. Review Comments to the Author

Reviewer #1: Dear Author,

First of all, I congratulate you on the study carried out. This is a relevant topic and your publication can contribute to a greater understanding of the literature. The text is written in a clear and concise manner, organized and complete in accordance with good scientific writing practices.

I enclose the reviewed manuscript with some suggested adjustments markings. Here I highlight three topics of greatest importance.

First, the unit of analysis of the study is not clear. From the shared database, it is supposed that you worked with a single unit of spatial analysis, the Federal District itself. This information must be in the Methods section.

The second topic refers to the methodological decision to use proportional mortality. For this purpose, the use of the odds ratio (OR) statistic was appropriate, both for performing the logarithmic regression and for the association test without knowledge of incidences. However, at different moments of the discussion and conclusion of the work, these statistics are interpreted as risk, which is not appropriate. I suggest revising the text and correcting all passages that make this mention.

Finally, I suggest further detailing the work limitations section, especially for the topics:

- it is an ecological study, therefore, subject to ecological bias for all relationships studied;

- depending on the size of the analysis unit, the statistical analysis may have lost power.

- use of proportional mortality does not allow conclusions regarding the risk of death, and other studies with adequate methodology are necessary to reach this conclusion.

I wish you success with the publication. Best regards.

6. PLOS authors have the option to publish the peer review history of their article (what does this mean?). If published, this will include your full peer review and any attached files.

Reviewer #1: **Yes: **Marcelo Pellizzaro Dias Afonso

---

## [Author Response · Author response to Decision Letter 0]

29 Apr 2022

Response to Reviewers 

Thank you for reviewing our article entitled “FACTORS ASSOCIATED WITH MORTALITY OF THE ELDERLY DUE TO AMBULATORY CARE SENSITIVE CONDITIONS, BETWEEN 2008 AND 2018, IN THE FEDERAL DISTRICT, BRAZIL”. The suggested changes were made and certainly helped substantially improve our work.

The answers are below:

1) The database of deaths belonging to the Government Health Department of the Federal District was used and monitored by the technical team of the Epidemiological Surveillance. It is a robust database that has been using the same software since 1998. Technicians evaluate the death certificates received, and if the data are inconsistent, additional information is requested from the hospitals or health units where the declarations were filled out. In addition, the Federal District has Death Verification Services (DVS) where professionals evaluate bodies whose cause of death is not well understood to define the cause of death. DVS performs autopsies in the following situations: diseases with mandatory notification, cases without definition of the underlying cause of death, and deaths in less than 24 hours of hospital admission, without diagnosis. The technicians for the death database of the Federal District meet weekly to ensure completeness of the data.

2) The choice of variables occurred after reviewing the database, correlating with the objectives of the article. Initially, the classic variables, such as sex, age, marital status, and race/color, were analyzed. We list the causes of death and indicate those that occurred due to conditions sensitive to ambulatory care, according to Ordinance No. 221 of April 17, 2008, by the Ministry of Health of Brazil. To correlate with primary care, we investigated the coverage by the health service, differentiating between the traditional primary care model and the family health strategy model according to the address of the deceased elderly person. We wanted to understand the importance of social inequalities in the deaths of the elderly. Once the deaths of the elderly in the Federal District were studied, different socioeconomic realities in the same geographic unit were analyzed. There are elderly people with high economic power, living in the best urban conditions, and very poor elderly people living in places with poor infrastructure. To understand the influence of socioeconomic factors, we correlated it with the Urban Well-Being Index (UWBI). 

This index is composed of five dimensions: mobility, environmental, and housing conditions, infrastructure, and collective services. Each of these dimensions is calculated, and then the arithmetic mean is calculated. The result is a value that ranges from 0 to 1. The UWBI is better the closer it is to 1 and worse the closer it is to 0. The address of the deceased elderly person was considered. We reiterate that all analyzed variables are plausible and previously had a possible causal relationship.

As for the expected results, the primary researcher, who is a family and community physician and works in the poorest region of the Federal District, wanted, among other things, to understand how the conditions of the place of residence influence the mortality of the elderly. We hypothesized that the analysis would show that the inequalities, expressed in our article by the UWBI, impact the mortality of the elderly, as the main researcher has empirically observed this phenomenon in his routine of care.

3) We chose to use the ecological variables primary care coverage and UWBI in our analysis. These variables were shown to be adequate for the hierarchical modeling of the article, since they analyze the context of the geographic, urban, and housing conditions of individuals. We indicate as a possible limitation of the study that this modeling can be impacted by the ecological bias. These variables in the article do not work as a period-time control.

4) The work performed a hierarchical modeling. In this model, we used age at the distal level (block A). The age effect was tested at this level, and in fact showed an association between older age and higher mortality from conditions sensitive to ambulatory care (OR: 1.775, p<0.001). The age variable was used to adjust the following variables at the medial level (block B) and at the proximal level (block C) to ensure better reliability.

Our study analyzed a common outcome for all research participants, the death of the elderly. To answer the hypotheses of the study, the outcome was classified as a binary variable: death from a sensitive condition – yes or no. The categorization of the outcome variable as binary would make the modeling resulting from counting unfeasible. Thus, we believe that the hierarchical model is more adequate for our objective.

Excellent colocation by the reviewer; however, the modeling performed answered our research question. Nevertheless, if the analysis suggested by the editor is essential, we would be willing to carry it out.

5) We made the change suggested by the editor, and we changed it to chance (odds). 

We await the final answer and are available for any clarifications.

Sincerely.

The authors.

---

## [Decision Letter · Decision Letter 1]

8 Jun 2022

PONE-D-21-36313R1FACTORS ASSOCIATED WITH MORTALITY OF THE ELDERLY DUE TO AMBULATORY CARE SENSITIVE CONDITIONS, BETWEEN 2008 AND 2018, IN THE FEDERAL DISTRICT, BRAZILPLOS ONE

Dear Dr. Marques da Costa,

Thank you for submitting your manuscript to PLOS ONE. After careful consideration, we feel that it has merit but does not fully meet PLOS ONE’s publication criteria as it currently stands. Therefore, we invite you to submit a revised version of the manuscript that addresses the points raised during the review process.

I still have two questions that were not addressed in the paper and I believe the discussion and contribution would benefit from it:Coverage and urban well-being do not vary by individual, they only vary by period of time. What is the limitation of doing this? Does it act as a control for period? Mortality increases with age - there are more deaths at older ages than any other age. Thus, it is expect that we see deaths increasing with your age in the analysis. One alternative is to model age specific mortality rates. One can obtain population by sex and age from IBGE or other sources. Then, you could model death rates or apply a poisson model on the counts of death. I believe modeling only death counts without a exposure measure limits the discussion of the results and might be affecting your results. Please submit your revised manuscript by Jul 23 2022 11:59PM. If you will need more time than this to complete your revisions, please reply to this message or contact the journal office at plosone@plos.org. Please include the following items when submitting your revised manuscript:A rebuttal letter that responds to each point raised by the academic editor and reviewer(s). You should upload this letter as a separate file labeled 'Response to Reviewers'.A marked-up copy of your manuscript that highlights changes made to the original version. You should upload this as a separate file labeled 'Revised Manuscript with Track Changes'.An unmarked version of your revised paper without tracked changes. You should upload this as a separate file labeled 'Manuscript'.If applicable, we recommend that you deposit your laboratory protocols in protocols.io to enhance the reproducibility of your results. Protocols.io assigns your protocol its own identifier (DOI) so that it can be cited independently in the future. For instructions see: https://journals.plos.org/plosone/s/submission-guidelines#loc-laboratory-protocols. Additionally, PLOS ONE offers an option for publishing peer-reviewed Lab Protocol articles, which describe protocols hosted on protocols.io. Read more information on sharing protocols at https://plos.org/protocols?utm_medium=editorial-email&utm_source=authorletters&utm_campaign=protocols.

We look forward to receiving your revised manuscript.

Kind regards,

Bernardo Lanza Queiroz, Ph.D

Academic Editor

PLOS ONE

Journal Requirements:

Reviewers' comments:

Reviewer's Responses to Questions

**Comments to the Author**

1. If the authors have adequately addressed your comments raised in a previous round of review and you feel that this manuscript is now acceptable for publication, you may indicate that here to bypass the “Comments to the Author” section, enter your conflict of interest statement in the “Confidential to Editor” section, and submit your "Accept" recommendation.

Reviewer #1: All comments have been addressed

2. Is the manuscript technically sound, and do the data support the conclusions?

Reviewer #1: Yes

3. Has the statistical analysis been performed appropriately and rigorously? 

Reviewer #1: Yes

4. Have the authors made all data underlying the findings in their manuscript fully available?

Reviewer #1: Yes

5. Is the manuscript presented in an intelligible fashion and written in standard English?

Reviewer #1: Yes

6. Review Comments to the Author

Reviewer #1: Dear authors,

Thank you for your attention and for the adjustments in your manuscript.

I wish you success with the publication.

Best regards.

7. PLOS authors have the option to publish the peer review history of their article (what does this mean?). If published, this will include your full peer review and any attached files.

Reviewer #1: **Yes: **Marcelo Pellizzaro Dias Afonso

---

## [Author Response · Author response to Decision Letter 1]

4 Jul 2022

Response to Reviewers

Dear reviewers

Thank you for reviewing our article entitled “FACTORS ASSOCIATED WITH MORTALITY OF THE ELDERLY DUE TO AMBULATORY CARE SENSITIVE CONDITIONS, BETWEEN 2008 AND 2018, IN THE FEDERAL DISTRICT, BRAZIL”. PONE-D-21-36313R1 The suggested changes were answered and definitely contributed to substantially improve our work.

The answers are as follows:

1- Coverage and urban well-being do not vary by individual, they only vary by period of time. What is the limitation of doing this? Does it act as a control for period? 

We appreciate the comment. We do not see any major limitation in the use of variables that may have changed over the period covered by the study, since for each death we associate what we believe to be the most appropriate value for that point in time. Well-being has only one measure for the period, so we had no choice of which value to use for each event. Primary Care coverage was defined for each death considering the address where the elderly person resided, who was covered by either Traditional Primary Care or Family Health Strategy in December of each year of the study period. This study was cross-sectional, and the variables mentioned were not included in order to control for the period, but rather to assess whether changes in their levels lead to an increase or decrease in the chance that the outcome of interest occurs. As it is not possible at the ecological level to have different values of these factors at the same moment in time, the only way to assess their effects was to analyze the set of deaths over several years, within the proposed methodological design.

2- Mortality increases with age - there are more deaths at older ages than any other age. Thus, it is expect that we see deaths increasing with your age in the analysis. One alternative is to model age specific mortality rates. One can obtain population by sex and age from IBGE or other sources. Then, you could model death rates or apply a poisson model on the counts of death. I believe modeling only death counts without a exposure measure limits the discussion of the results and might be affecting your results. 

As noted by the reviewer, we believed that age could impact the chance of dying. To adjust for the effect of age, it was included in the distal level of the hierarchical model, which allows us to interpret all coefficients at the same level and at subsequent levels (intermediate and proximal) as independent effects of the variables, already adjusted for age. Our objectives were to investigate possible predictors of dying from ambulatory care sensitive conditions in the age group of the study. For this reason, we opted for logistic regression (binary outcome) with the hierarchical approach already mentioned.

As indicated in the answer to item 1, this is a cross-sectional study, and the correction is already explained in the materials and methods section.

We appreciate the considerations and are available to conduct any additional analyzes requested by the reviewer.

We await the final answer and are available for any clarifications.

Sincerely.

The authors.

---

## [Editor Report · Decision Letter 2]

25 Jul 2022

FACTORS ASSOCIATED WITH MORTALITY OF THE ELDERLY DUE TO AMBULATORY CARE SENSITIVE CONDITIONS, BETWEEN 2008 AND 2018, IN THE FEDERAL DISTRICT, BRAZIL

PONE-D-21-36313R2

Dear Dr. Marques da Costa,

We’re pleased to inform you that your manuscript has been judged scientifically suitable for publication and will be formally accepted for publication once it meets all outstanding technical requirements.

Kind regards,

Edward Jay Trapido, ScD

Academic Editor

PLOS ONE
---

## [Editor Report · Acceptance letter]

28 Jul 2022

PONE-D-21-36313R2 

Factors associated with mortality of the elderly due to ambulatory care sensitive conditions, between 2008 and 2018, in the Federal District, Brazil 

Dear Dr. Marques da Costa:

I'm pleased to inform you that your manuscript has been deemed suitable for publication in PLOS ONE. Congratulations! Your manuscript is now with our production department. 

Kind regards, 

on behalf of

Dr. Edward Jay Trapido 

Academic Editor

PLOS ONE